# Winter Weather-Related Crashes during the Commute to Work: An Opportunity for Total Worker Health^®^

**DOI:** 10.3390/ijerph181910268

**Published:** 2021-09-29

**Authors:** Jonathan Davis, Diane S. Rohlman

**Affiliations:** Department of Occupational and Environmental Health, University of Iowa, 145 North Riverside Drive, Iowa City, IA 52242, USA; diane-rohlman@uiowa.edu

**Keywords:** Total Worker Health, traffic safety, crash, wellness, commute

## Abstract

Background: The ways workers interface with their workplace and work are changing. These changes provide challenges and opportunities for health and safety professionals attempting to improve worker wellbeing for the future of work. For many workers, the morning commute, an activity typically outside the scope of workplace policies, is the most hazardous portion of the day. The hazard increases if workers are required to drive during winter weather or in other hazardous conditions in order to adhere to strict workplace start times. This research describes the role winter weather plays during the morning commute, demonstrating the need for organizational design and work arrangements that improve safety during the commute to work. Methods: Crash data from the Iowa Department of Transportation for the years 2013–2017 was linked to county level characteristics from the American Community Survey. Crashes were characterized by 30-min time intervals. The likelihood of the crash involving winter weather as a contributing circumstance was compared across time-intervals. Results: Winter weather was more likely to contribute to crashes during the commuting hours compared to 11:00 to 11:59 am. Winter weather was most frequently a contributing circumstance during 8:00–8:29 a.m. (OR = 2.21 95% CI: 1.93–2.52). Conclusions: Winter weather plays a role in crashes during commuting hours. Workplaces can adopt policies for flexible work start times or for telecommuting to empower workers to avoid hazardous driving conditions.

## 1. Introduction

Workers are often required to commute to work and arrive at a specific start time. Many workers lack the flexibility of deciding when they start their commute to work. While drivers do choose to delay or forgo driving in poor weather conditions [1,2], reduced driving is less apparent during the peak driving hours of 6:00 to 9:00 a.m., the common time period for the daily commute to work [1]. The commute to work is a significant source of employee injury, with motor vehicle crashes occurring most frequently during the morning and evening hours [3,4]. Although injury during a commute to work is not considered a work-related injury in the United States (OSHA 29 CFR 1904.5(b)(2)), the National Institute for Occupational Safety and Health (NIOSH) has expanded current injury and illness prevention efforts to non-occupational sources of increased risk in order to promote overall worker wellbeing through the concept of Total Worker Health^®^ (TWH) [5].

Approaches to TWH have evolved to include issues related to commuting, such as reducing stress and injury prevention [6]. Workplace policies and research about commuting have focused on driving after shift work [7], the stress caused by long commutes [8], and encouraging active travel [9,10,11,12]. Previous studies have not evaluated how flexible schedules or allowances for working remotely (telecommuting, alternative work arrangements) can impact driver safety. Recognizing the evolving relationship between workers and their workplace, NIOSH has developed priority topics through the Future of Work Initiative [13]. The commute to work is directly related to several of the topics and subtopics outlined in the initiative. Most directly, the safety of the commute can be improved through organizational design encouraging autonomy, stress prevention, and job flexibility, as well as through work arrangements that reconceptualize where and when work is completed. While the commute itself is not a job task and not the responsibility of the employer, the employer can directly affect the safety of the commute using the principles outlined in TWH and the Future of Work. 

A renewed focus on worker wellbeing through job flexibility and alternative work arrangements should include consideration for reducing traveling to work during adverse weather conditions. Adverse weather conditions contribute to 16% of all fatal crashes in the United States [14]. Analysis matching time-periods of snowfall with non-snowfall periods [15,16,17] and evaluation of increasing snowfall [18,19] have found that snowfall increases the rate of motor vehicle crashes. With a typical commute for US drivers lasting 27 min [20], many opportunities exist for an adverse weather-related related crash during the commute. 

In this study, we evaluated the likelihood that a crash was winter weather-related throughout the morning commuting hours. Using state crash data, we separated the time of crash into 30-min intervals and calculated the odds of winter weather being a reason for the crash. We hypothesized that winter weather is more likely a factor in crashes during the most common commute times compared to times with less commuter traffic. 

## 2. Materials and Methods

### 2.1. Study Design and Data Sources 

Iowa Department of Transportation (DOT) crash data was used to identify all crashes occurring in Iowa between midnight (12:00 a.m.) and 11:59 a.m. for drivers aged 16 through 64 during the years of 2013 to 2017. The Iowa DOT crash data is publicly available as the Iowa Crash Analysis Tool [21]. The tool provides information on all crashes that resulted in an injury or fatality or resulted in estimated property damage of $1500 or more. Information about morning commute activity at the county level is also publicly available in the American Community Survey (ACS) [22]. The ACS was used to identify characteristics of a commute at the county level for each year. Conditions identified included mean travel time, traffic density during specific times, and the total number of drivers commuting. Traffic density was derived from the ACS question asking commuters what time they typically left for work. Additionally, the number of total drivers was determined as those who indicated that they typically drive to work. These two measures together describe how many drivers are expected to be on the road at a given time. These conditions were linked to the crash data based on the county and time of the crash.

We used a cross-sectional design to identify the increased odds of winter weather contributing to the crash (winter weather-related) throughout various times during the morning commute with 11:00 to 11:59 a.m. as a referent category. 

The analysis of these publicly available datasets was reviewed by the University of Iowa Institutional Review Board and determined to not be human subject research. 

### 2.2. Winter Weather-Related Crash and Any Weather-Related Crash

A winter weather-related crash was defined as a crash with any of the following: a contributing circumstance of “wet, icy”; surface conditions of “blowing snow,” “freezing rain/drizzle,” “sleet, hail,” or “snow”; surface conditions of “ice/frost,” “snow,” or “slush”. The days between 23 December and 2 January were excluded to remove days with uncharacteristic commute conditions due to anticipated winter vacations. We also identified any weather-related crash as defined as a crash with a contributing circumstance of “Weather conditions” listed in the environmental contributing circumstances field of the crash data. The analysis of any weather-related crash used all months during the year. 

### 2.3. Crash Characteristics and Injury Severity

Crash characteristics were identified from the Iowa DOT crash data. The following crash characteristics were collected and described in this analysis: day of the week, road system, speed limit, paved road, and drug- or alcohol-related. The road system variable had the following classifications: construction (construction zone), farm to market (route to allow flow of commodities from rural areas to towns and terminals), interstate (highway), local road (public roads to assure connection of parcels of land), Iowa route, and US routes. The speed limit variable was the statuary or posted maximum speed for the area in miles per hour (MPH). Unpaved roads included gravel and dirt roads as opposed to paved surfaces. 

Injury characteristics were also described using the Iowa DOT crash data. Whether or not any injury occurred as a result of the crash, multicar crash (one or more cars), and the amount of property damage were described for drivers with and without winter weather-related crashes. 

### 2.4. Data Analysis

Odds ratios and 95% confidence intervals were calculated for the increased odds of a crash being winter weather or any weather-related during the morning commute time categories (12:00–04:59 a.m., 5:00–5:29 a.m., 5:30–5:59 a.m., 6:00–6:29 a.m., 6:30–6:59 a.m., 7:00–7:29 a.m., 7:30–7:59 a.m., 8:00–8:29 a.m., 8:30–8:59 a.m., 9:00–9:29 a.m., 9:30–9:59 a.m., 10:00–10:29 a.m., 10:30–10:59 a.m.) compared to 11:00–11:59 a.m. The odds of a winter weather-related crash to non-winter weather-related crash occurring during the times categories was compared with the odds of a winter weather-related crash to non-winter weather-related crash occurring during 11:00 a.m. to 11:59 a.m. using logistic regression. The same methods were used for any weather-related crash. For both winter weather-related crashes and any weather-related crash, we only included crashes occurring during Monday–Friday to reflect the typical work week. For the analysis of winter weather-related crashes, analysis was restricted to the months of November to February. We also evaluated the odds ratio of winter weather-related and any weather-related crashes occurring for female drivers compared to male drivers and across age groups using logistic regression. Conditions of the commute were controlled for in the logistic model by including variables of mean travel time, traffic density, and the total number of commuting drivers in the county where the crash occurred. Characteristics of winter weather-related crashes were compared to characteristics of all other crashes using the Chi-square test. For categorical variables with more than two levels, the presence of the characteristic was compared to the absence of that characteristic across each level of the category. All tests were completed with a significance level of α = 0.05.

## 3. Results

### 3.1. Comparison of Crash Characteristics

During relevant time-periods and days from 2013 to 2017, there were 85,781 total crashes. Of these crashes, 19.1% met our criteria for winter weather-related crashes. The differences between winter weather and non-winter weather-related crashes are given in Table 1. Winter weather was most frequently a contributing cause of a crash on Monday (*n* = 2973, 18.2%) and Tuesday (*n* = 2961, 18.1%). Winter weather-related crashes were less likely to occur for the days Thursday through Saturday. For Sundays, there was no difference in the likelihood of the crash being winter weather-related or non-winter weather-related. The type of road system where the crash occurred was significantly different between winter weather and non-winter weather crashes for all types of road systems except for construction zones. The biggest difference was for crashes occurring on the interstate, where 18.3% of winter weather-related crashes occurred compared to 12.0% of other crashes (*p* < 0.001). The only other road system with a higher proportion of crashes being winter weather-related was farm to market routes (*p* < 0.001). 

The speed limit for the location of the crash was different for all speed limits except for the ≤25 MPH across winter weather-relatedness (Table 1). Winter weather-related crashes were more likely to occur for all speed limits above 50 MPH compared to other crashes. Crashes were less likely to have drug or alcohol involvement if winter weather contributed to the crash (3.1% winter weather-related vs. 7.4% non-winter weather-related, Table 1). The drug or alcohol-related winter weather crashes included 18.6% (*n* = 93) who refused the sobriety test. A similar proportion (19.3%, *n* = 990) of the drug or alcohol-related crashes occurring without winter weather conditions had drivers refusing to test. 

### 3.2. Difference in Winter Weather-Related Crashes across Commuting Times

The odds ratios for a crash occurring during a given time compared to the time of 11:00 a.m. to 11:59 a.m. are shown in Table 2. Except for the earliest morning hours (00:00 midnight to 04:59 a.m.) all periods were significantly more likely to have a winter weather-related crash. The distribution of crash odds ratios was bimodal, with crash odds peaking during 5:30–5:59 a.m. (OR = 1.62 95% CI: 1.37–1.91) and 8:00–8:29 a.m. (OR = 2.21 95% CI: 1.93–2.52) compared to 11:00–11:59 am. Analysis was also completed for crashes with any weather-related cause throughout the entire year. The odds ratio for an any weather-related crash remained elevated through the morning hours, but began to taper after 8:00–8:29 a.m. (OR = 2.24 95% CI: 1.96–2.57). From the ACS survey results, 79% of Iowans reported beginning their commute before 9:00 a.m. Of the half-hour commute periods asked about in the ACS, drivers most frequently reported leaving between 7:30 and 8:00 a.m. (15.4%, data not shown). There was no difference in the odds of a winter weather-related crash for female drivers compared to male drivers. However, in the analysis of the entire year, female drivers were less likely to be involved in an any weather-related crash. There was an overall decrease in the odds of a winter weather-related crash for the three age groups of drivers over the age of 35. The lowest odds of a winter weather-related crash were calculated for drivers in the oldest age group, that being 55–64 years old. Drivers aged 55–64 were also less likely to be involved in an any weather-related crash compared to 16 to 24 year-old drivers. 

### 3.3. Injury and Crash Severity

The severity of winter weather-related crashes was compared with all other crashes (Table 3). Winter weather-related crashes had lower fatality (0.4% vs. 0.7%, *p* < 0.001) and were less likely to result in any injury (24.0% vs. 27.4%, *p* < 0.001) when compared to other crashes. This may be partially explained by weather-related crashes less frequently resulting in a multicar crash compared to other crashes (54.2% vs. 61.9%, *p* < 0.001). Winter weather-related and non-winter weather-related crashes resulted in similar amounts of property damage. 

## 4. Discussion

We found that the likelihood of winter weather-related crashes was higher during the commuting hours compared to 11:00 to 11:59 a.m. We examined how this increased risk changes throughout the morning. The role of adverse winter weather conditions in crashes did not subside until well after most people would have completed their morning commute. Winter weather was most likely to play a role in crashes occurring between 7:30 and 9:00 a.m. While it was uncommon for drivers to begin their commute after 9:00 a.m., winter weather-related crashes were more likely during the commute times between 9:00 and 11:00 a.m. compared to the 11:00 to 11:59 a.m. time period. 

An important mechanism to reduce the likelihood of winter weather-related crashes during the commute is to allow more flexibility for when workers are required to physically be at work. Through alternative work arrangements or flexible work hours, employers can reduce the frequency of employees driving in poor conditions. Additionally, flexible work schedules reduce the need for a driver to be on the road at specific times, allowing for the improvement of the surface conditions through plowing and application of anti-icing agents. Flexible work hours and alternative work arrangements are subtopics of the Future of Work Initiative [13]. In addition to the many ways that these programs can improve worker wellbeing, as outlined in the initiative, the reduced stress from driving in adverse conditions and the improved safety from reduced crash risk further justifies adoption of these programs.

Several studies support that proposition that driving in snowy conditions increases the risk of a crash [15,16,17,18,19,23]. Despite the known risk, there is a lack of guidance on how to reduce injuries to workers occurring during their commute, and no evaluations of policies that allow for flexible work start times or required work from home when adverse conditions are present. Other risk factors for a crash that can be influenced by workplace polices have been evaluated. Shift work and extended work shifts have both been identified as increasing the risk for a crash [7,24]. These risk factors can be mitigated through policies requiring a minimum amount of time between shifts and limiting the number of hours or number of extended shifts. Similarly, the increased risk from driving in adverse conditions can be reduced through policies allowing for alternative work arrangements. 

Workplace programs such as flexible scheduling options and telecommuting are less likely to be available for employees of low-paying jobs [25,26]. Programs that allow for flexible start times or the option to work from home when adverse weather conditions are present are likely to have similar disparities in application. Many low-paying jobs in sectors such as customer service, retail, manufacturing, and food service have set time periods requiring employees to be present onsite to complete job requirements. Employers of these jobs may find it more difficult to implement policies for flexible start times. However, our work provides evidence of the risk of crash resulting from driving in poor weather conditions. Implementing flexible start time and telecommuting programs could help reduce this risk. The benefits of these programs should be considered even if implementation is difficult. Other studies of work commute have shown that changes in the commute time can reduce job strain and burnout [27,28]. Improved job satisfaction should be evaluated in future studies of flexible start time and telecommuting programs as additional benefits beyond the reduced risk of injury from motor vehicle crashes.

Winter weather-related crashes more frequently occurred at the beginning of the workweek and on high speed, interstate roadways. These crashes were less likely to occur with a report of alcohol or drug involvement. Similarly, a study of the Fatality Analysis Reporting System (FARS), a national system that collects data about crashes that occur on public roads in the US, found that fatal crashes that occurred with an adverse weather event were less likely to involve drug or alcohol use compared to other fatal crashes [14]. The decreased contribution of alcohol and drugs in winter weather-related crashes suggests modification of these risk factors alone will not sufficiently improve driving safety in poor conditions.

Road conditions are a major factor in winter weather-related crashes. Methods to improve conditions, such as salting, sanding, and plowing have been shown to decrease the risk of motor vehicle crashes [29]. Based on the road system, improving surface conditions can increase road safety [30]. We found that a larger proportion of winter weather-related crashes occurred on interstates compared to other types of crashes. Improvements to surface conditions and road safety features of interstates should be prioritized in areas with many commuters and frequent winter weather conditions. 

In previous research, female drivers have been identified as having a higher likelihood of being in an injurious crash during commuting hours [4]. We found that female drivers and male drivers had a similar likelihood of having a winter weather-related crash. However, male drivers did have a higher likelihood of fatal winter weather-related crashes than female drivers. The similar winter weather-related crash risk suggests female drivers and male drivers are facing similar demands to drive in adverse winter weather-related conditions. The likelihood of winter weather-related crash was lower for older age groups. We suspect these drivers have more flexibility in their work start time and are more frequently avoiding driving in adverse winter weather conditions. The assumed increased flexibility of these older drivers may change with a shift in the underlying worker population demographics. The job requirements of current older workers may differ from the job requirements of future older workers as they make up a larger proportion of the workforce. This change may limit older worker flexibility in the future if employers don’t adopt flexible work arrangements. Additional research about work start time flexibility across age groups is needed to further understand the potential impact to further reduce crash risk. 

Winter weather-related crashes, in our study, were less likely to result in injuries or death compared to non-winter weather-related crashes. In a previous study of Iowa interstate highway crashes during a snow event, crashes were less likely to result in an injury when compared to crashes occurring during non-snow events [31]. Multi-car crashes were less frequent among the winter weather-related crashes in our study. We believe this played a role in crashes being less severe since only one car was involved.

A limitation of this study is our inability to determine traffic flow during snowfall. Traffic flow decreases during hours of heavy snowfall [1,2]. However, decreased traffic flow has been shown to be less impacted at peak hours, such as during the morning commute [1]. This is supported by our findings of an over-representation of winter weather-related crashes occurring during most common commute times. Another limitation was our reliance on county-level data to control for differences in traffic density, the total number of drivers, and travel time. This may not accurately reflect the conditions at the time of the crash. However, it allowed us to generally identify driving conditions that may have contributed to the crash. Additionally, the use of county-level data provides an example of how the ACS can be used at the ecological level to control for different driving conditions. This study only provides suggestive evidence of an over-representation of winter weather-related crashes occurring during commute hours. The setting for this study is a midwestern state with heavy snowfall. The specific weather conditions of a location should be considered when creating programs that address flexibility around the commute to work. Our secondary analysis found an increased likelihood of any weather-related crash during the commute hours, so we suggest other weather conditions be included in policies allowing for flexible works schedules and alternative work arrangements. Most commuters in Iowa do not use public transportation to travel to work. Safe and affordable public transportation systems can allow drivers to avoid driving in hazardous conditions. If workplaces are in areas with public transportation infrastructure, employers and safety managers could support employee use of these modes of transportation as another means of reducing the worker exposure to hazardous driving conditions. TWH research of workplace programs should include organizational and worker- level measurements to fully understand the impact of such programs [32]. More comprehensive research is needed to determine the impact of specific workplace programs that allow for flexible start times and alternative work arrangement to reduce driving in hazardous conditions. 

## 5. Conclusions

We found that drivers were more likely to be involved in a winter weather-related crash during typical commute times. If drivers are given greater flexibility to avoid driving during adverse conditions, weather-related crashes could be reduced. Current commuting practices of self-restriction of driving are not mitigating the risk posed by driving in adverse weather conditions. An emphasis for employers and researchers tasked with improving the safety of a workplace commute has been reducing drowsy driving for shift work employees [7]. The reduction of drowsy driving through workplace policy is an example of how TWH principals can be applied to a driver’s commute. Even more drivers would potentially be affected by policies allowing for flexible travel times or working remotely, as most workers are not exposed to shift work. While not possible for all workplaces, the dangers of a commute should be considered whenever possible to improve the overall health of employees. States can use crash data along with the ACS survey to assess how well drivers are avoiding hazardous driving conditions and if weather-related crashes are overrepresented during typical work commute hours.

## Figures and Tables

**Table 1 ijerph-18-10268-t001:** Crash characteristics for crashes with and without a winter weather-related circumstance.

Variable	Category	All Other Crash		Winter Weather-Related Crash		*p*-Value ^1^
		*n* = 69,403	%	*n* = 16,378	%	
Day of Week	Monday	10,162	14.6	2973	18.2	<0.001
	Tuesday	10,318	14.9	2961	18.1	<0.001
	Wednesday	10,144	14.6	2698	16.5	<0.001
	Thursday	10,481	15.1	1853	11.3	<0.001
	Friday	10,957	15.8	2207	13.5	<0.001
	Saturday	9655	13.9	1906	11.6	<0.001
	Sunday	7686	11.1	1780	10.9	0.449
Road System	Construction	257	0.4	62	0.4	0.876
	Farm to Market Route	13,957	20.1	3520	21.5	<0.001
	Interstate	8308	12.0	2995	18.3	<0.001
	Iowa Route	9335	13.5	1615	9.9	<0.001
	Local Road	24,694	35.6	5394	32.9	<0.001
	US Route	12,852	18.5	2792	17.0	<0.001
Speed Limit	≤25 MPH	16,898	28.5	3927	24.4	0.320
	30–35 MPH	17,131	28.9	3074	19.1	<0.001
	40–45 MPH	5804	9.8	1300	8.1	0.076
	50–55 MPH	13,214	22.3	4308	26.7	<0.001
	60–65 MPH	4002	6.7	2093	13.0	<0.001
	≥70 MPH	2270	3.8	1403	8.7	<0.001
Paved	Paved	65,899	95.3		96.1	
	Unpaved	3247	4.7	642	3.9	<0.001
Drug or Alcohol Related	No	64,276	92.6	15,877	96.9	
	Yes	5127	7.4	501	3.1	<0.001

^1^ The *p*-value is calculated using the chi-square test.

**Table 2 ijerph-18-10268-t002:** Odds Ratios for time period having a winter weather-related and any weather-related circumstance during the work week.

Variable	Winter Weather-Related Crash ^1^			Any Weather-Related Crash ^2^	
	Case/Control ^3^	OR ^4^	95% CI	OR ^4^	95% CI
Time of Day					
00:00–04:59	893/1994	0.78	(0.67, 0.90)	1.11	(0.97, 1.28)
05:00–05:29	207/312	1.34	(1.10, 1.63)	1.99	(1.68, 2.36)
05:30–05:59	362/488	1.62	(1.37, 1.91)	2.62	(2.25, 3.05)
06:00–06:29	389/740	1.25	(1.07, 1.48)	2.22	(1.89, 2.61)
06:30–06:59	652/1147	1.51	(1.26, 1.80)	2.65	(2.21, 3.16)
07:00–07:29	752/1194	1.75	(1.46, 2.11)	2.35	(1.95, 2.83)
07:30–07:59	1291/1861	1.91	(1.60, 2.28)	2.12	(1.77, 2.54)
08:00–08:29	1238/1350	2.21	(1.93, 2.52)	2.24	(1.96, 2.57)
08:30–08:59	857/966	1.92	(1.70, 2.17)	1.83	(1.62, 2.08)
09:00–09:29	619/732	1.81	(1.58, 2.06)	1.61	(1.41, 1.85)
09:30–09:59	588/748	1.67	(1.46, 1.91)	1.52	(1.32, 1.74)
10:00–10:29	557/817	1.44	(1.26, 1.65)	1.44	(1.26, 1.66)
10:30–10:59	479/826	1.24	(1.08, 1.42)	1.25	(1.09, 1.45)
11:00–11:59	927/1976	Ref		Ref	
Female	4043/6286	0.98	(0.93, 1.03)	0.87	(0.82, 0.91)
Male	5396/8395	Ref		Ref	
Age					
16–24	2696/3684	Ref		Ref	
25–34	2459/3628	0.94	(0.88, 1.02)	1.03	(0.96, 1.11)
35–44	1773/2798	0.87	(0.80, 0.94)	0.93	(0.86, 1.00)
45–54	1613/2682	0.82	(0.76, 0.89)	0.93	(0.86, 1.01)
55–64	1270/2359	0.73	(0.67, 0.79)	0.81	(0.74, 0.88)

^1^ A winter weather-related crash was defined as a crash with any of the following: a contributing circumstance of “wet, icy”; surface conditions of “blowing snow,” “freezing rain/drizzle,” “sleet, hail,” or “snow”; surface conditions of “ice/frost,” “snow,” or “slush”. Analysis was restricted to November through February and the typical work week of Monday-Friday. ^2^ Any weather-related crash as defined as a crash with a contributing circumstance of “Weather conditions.” Analysis was restricted to Monday-Friday. ^3^ Case was defined as a winter weather-related crash. Crashes that were not winter weather-related were controls. ^4^ Model includes county level categorical variables for mean travel time, traffic density, and the total number of commuting drivers.

**Table 3 ijerph-18-10268-t003:** Crash severity for crashes with and without a winter weather-related circumstance.

Variable	Category	All Other Crash		Winter Weather-Related Crash		*p*-Value ^1^
		*n* = 69,403	%	*n* = 16,378	%	
Crash Severity	Fatal Crash	514	0.7	60	0.4	<0.001
	Possible/Unknown Injury Crash	10,758	15.5	2405	14.7	0.009
	Property Damage Only	50,056	72.1	12,412	75.8	<0.001
	Suspected Minor Injury Crash	6364	9.2	1228	7.5	<0.001
	Suspected Serious Injury Crash	1711	2.5	273	1.7	<0.001
Any Injury	No Injury	50,369	72.6	12,445	76.0	
	Injury	19,034	27.4	3933	24.0	<0.001
Multicar Crash	No	26,424	38.1	7499	45.8	
	Yes	42,979	61.9	8879	54.2	<0.001
Property Damage	Median	4500	-	4500	-	

^1^ The *p*-value is calculated using the chi-square test.

## Data Availability

Publicly available datasets were analyzed in this study. This data can be found here: https://icat.iowadot.gov/ (accessed on 10 May 2020) and https://www.census.gov/acs/www/data/data-tables-and-tools/subject-tables/ (accessed on 10 May 2020).

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
