# Peer review of "Winter Weather-Related Crashes during the Commute to Work: An Opportunity for Total Worker Health®"

_ijerph, 2021, doi:10.3390/ijerph181910268_

Round 1
Reviewer 1 Report
This paper analyzes automobile crashes in half-hour increments during the morning hours and compares them to 11:00-11:59 a.m. to determine the crash risk associated with various times of going to work during winter weather conditions. This research uses data from the Iowa Department of Transportation crashes for the years 2013 to 2017 and has developed some interesting research findings. In general, the paper is well-written and persuasively argued. However, the results would be more robust if they considered (and controlled for) the number of vehicles on the roadway at specific times of the morning using either a measure of average annual daily traffic (AADT) or some other factor to account for the fact that more activity on the highways is likely to be associated with a greater risk of collisions (at least collisions with other vehicles). The authors may be able to develop surrogates for this data if AADT is not available on enough roads (e.g., they might separate out single-vehicle crashes from other collisions). One other consideration as the authors revise this paper is that weather-related crashes may have different characteristics than non-weather related crashes. For example, crashes involving multiple vehicles may be more likely to be weather-related in areas with more traffic while single-vehicle collisions (with a snow-bank or roadway infrastructure (e.g., guardrails) may be indicative of a weather-related crash. These relationships could be sorted out using machine learning or other techniques but even with this methodology, the authors could use additional information that is included in the crash reporting system to control for some of the variability in the data.
Author Response
This paper analyzes automobile crashes in half-hour increments during the morning hours and compares them to 11:00-11:59 a.m. to determine the crash risk associated with various times of going to work during winter weather conditions. This research uses data from the Iowa Department of Transportation crashes for the years 2013 to 2017 and has developed some interesting research findings. In general, the paper is well-written and persuasively argued. However, the results would be more robust if they considered (and controlled for) the number of vehicles on the roadway at specific times of the morning using either a measure of average annual daily traffic (AADT) or some other factor to account for the fact that more activity on the highways is likely to be associated with a greater risk of collisions (at least collisions with other vehicles). The authors may be able to develop surrogates for this data if AADT is not available on enough roads (e.g., they might separate out single-vehicle crashes from other collisions).
Author Response: Thank you for the comments. We do control for traffic density using responses from the American Community Survey. We feel we did not adequately describe how these measures were derived. We have revised the methods section to clarify how we controlled for number of vehicles on the road.
Revision to manuscript:
“The ACS was used to identify characteristics of a commute at the county level for each year. Conditions identified included mean travel time, traffic density during specific times, and total drivers commuting. Traffic density was derived from the ACS question asking commuters what time they typically left for work. Additionally, the number of total drivers was determined as those who indicated they typically drive to work in the ACS survey. These two measures together provide a measure of how many drivers are on the road at a given time.”
One other consideration as the authors revise this paper is that weather-related crashes may have different characteristics than non-weather related crashes. For example, crashes involving multiple vehicles may be more likely to be weather-related in areas with more traffic while single-vehicle collisions (with a snow-bank or roadway infrastructure (e.g., guardrails) may be indicative of a weather-related crash. These relationships could be sorted out using machine learning or other techniques but even with this methodology, the authors could use additional information that is included in the crash reporting system to control for some of the variability in the data.
Author Response: For descriptive purposes we did choose to describe difference in injury and multicar involvement of winter and non-winter related crashes (Table 3). We did find winter weather-related crashes to more frequently be single vehicle (45.8% vs 38.1% non-winter weather). We also discussed some difference in road characteristics (speed limit, paved; road system, Table 1). While describing road infrastructure would additional context, the goal of research is to lessen winter-related crashes through flexible work schedules, so we focused on describing characteristics on a less granular scale thus encompassing more of the commute itself. Safety infrastructure is important but has been addressed in the traffic safety literature. There is no current research about how flexible start times could reduce crash risk. Our work provides some initial context for the impact flexible work schedules could have on crash risk.
In response to the suggestion of controlling for these variables, we do not believe statistical adjustment is appropriate for these characteristics. Our unit of analysis is time and road characteristics are fixed, so these characteristics do not confound our analysis i.e. guardrails exists at both 7:00 am and 11:00 am. They may be related to the crash but do not confound the relationship between commute time and likelihood of winter weather-relatedness.
Reviewer 2 Report
This manuscript addresses the winter weather-related crashes during commute times. In principle, the project is an interesting and important study. I have two concerns regarding the manuscript.
* The conclusions of this study are too simple. The authors should analyze the winter weather-related crashes from the perspective of traffic safety management.
* There are significant language shortcomings (for example: line 79, "The analysis of these publicly available datasets was review by the [anonymized for review] Institutional Review Board and determined to not be human subjects research").
Author Response
This manuscript addresses the winter weather-related crashes during commute times. In principle, the project is an interesting and important study. I have two concerns regarding the manuscript.
* The conclusions of this study are too simple. The authors should analyze the winter weather-related crashes from the perspective of traffic safety management.
Author Response: We have expanded our discussion to cover issues related to traffic safety management. The section beginning on line 232 now reads:
“Road conditions are a major factor in winter weather-related crashes. Methods to improve conditions, such as salting or sanding and plowing have been shown to decrease the risk of motor vehicle crashes [22]. Based on the road system, improving surface conditions can increase road safety [23]. We found that a larger proportion of winter weather-related crashes occurred on interstates compared to other types of crashes. Improvements to surface conditions and road safety features of interstates should be prioritized in areas with many commuters and frequent winter weather conditions.
An important mechanism to reduce the likelihood of winter weather-related crashes during the commute is to allow more flexibility for workers to choose when they leave for work to reduce the frequency of driving in poor conditions. This allows drivers to choose to drive when the weather conditions improve, as well as, allowing time for the improvement of the surface conditions through plowing and application of anti-icing agents.”
* There are significant language shortcomings (for example: line 79, "The analysis of these publicly available datasets was review by the [anonymized for review] Institutional Review Board and determined to not be human subjects research").
Author Response: We have made several language edits throughout the document.